# Biogeochemical processes create distinct isotopic fingerprints to track floodplain rearing of juvenile salmon

**Miranda Bell-Tilcock**[1]*, **Carson A. Jeffres**[1], **Andrew L. Rypel**[1,2], **Malte Willmes**[3,4], **Richard A. Armstrong**[5], **Peter Holden**[5], **Peter B. Moyle**[1], **Nann A. Fangue**[2], **Jacob V. E. Katz**[6], **Ted R. Sommer**[7], **J. Louise Conrad**[8], **Rachel C. Johnson**[1,4]

1 Center for Watershed Sciences, University of California, Davis, CA, United States of America, 2 Department of Wildlife, Fish & Conservation Biology, University of California, Davis, CA, United States of America, 3 Institute of Marine Sciences, UC Santa Cruz, Santa Cruz, CA, United States of America, 4 National Marine Fisheries Service, Southwest Fisheries Science Center, Santa Cruz, CA, United States of America, 5 Research School of Earth Sciences, Australian National University, Acton, ACT, Australia, 6 CalTrout, San Francisco, CA, United States of America, 7 Department of Water Resources, West Sacramento, CA, United States of America, 8 Delta Stewardship Council, Sacramento, CA, United States of America

* mirbell@ucdavis.edu

**Data Availability Statement:** Data is available at: Miranda Bell Tilcock. (2021). mirbell/CNS-isotopes-for-floodplains: Release CNS Floodplains

## Abstract

Floodplains represent critical nursery habitats for a variety of fish species due to their highly productive food webs, yet few tools exist to quantify the extent to which these habitats contribute to ecosystem-level production. Here we conducted a large-scale field experiment to characterize differences in food web composition and stable isotopes ($\delta^{13}C$, $\delta^{15}N$, $\delta^{34}S$) for salmon rearing on a large floodplain and adjacent river in the Central Valley, California, USA. The study covered variable hydrologic conditions including flooding (1999, 2017), average (2016), and drought (2012–2015). In addition, we determined incorporation rates and tissue fractionation between prey and muscle from fish held in enclosed locations (experimental fields, cages) at weekly intervals. Finally, we measured $\delta^{34}S$ in otoliths to test if these archival biominerals could be used to reconstruct floodplain use. Floodplain-reared salmon had a different diet composition and lower $\delta^{13}C$ and $\delta^{34}S$ ($\delta^{13}C$ = -33.02±2.66‰, $\delta^{34}S$ = -3.47±2.28‰; mean±1SD) compared to fish in the adjacent river ($\delta^{13}C$ = -28.37±1.84‰, $\delta^{34}S$ = +2.23±2.25‰). These isotopic differences between habitats persisted across years of extreme droughts and floods. Despite the different diet composition, $\delta^{15}N$ values from prey items on the floodplain ($\delta^{15}N$ = 7.19±1.22‰) and river ($\delta^{15}N$ = 7.25±1.46‰) were similar, suggesting similar trophic levels. The food web differences in $\delta^{13}C$ and $\delta^{34}S$ between habitats were also reflected in salmon muscle tissue, reaching equilibrium between 24–30 days (2014, $\delta^{13}C$ = -30.74±0.73‰, $\delta^{34}S$ = -4.6±0.68‰; 2016, $\delta^{13}C$ = -34.74 ±0.49‰, $\delta^{34}S$ = -5.18 ±0.46‰). $\delta^{34}S$ measured in sequential growth bands in otoliths recorded a weekly time-series of shifting diet inputs, with the outermost layers recording time spent on the floodplain ($\delta^{34}S$ = -5.60±0.16‰) and river ($\delta^{34}S$ = 3.73±0.98‰). Our results suggest that $\delta^{13}C$ and $\delta^{34}S$ can be used to differentiate floodplain and river rearing habitats used by native fishes, such as Chinook Salmon, across different hydrologic conditions and

Code (v1.0). Zenodo. https://doi.org/10.5281/zenodo.5514074.

**Funding:** USBR contract to Jacob: R13PS20147, DWR contract to UCD #4600011527.

**Competing interests:** The authors have declared that no competing interests exist.

tissues. Together these stable isotope analyses provide a toolset to quantify the role of floodplains as fish habitats.

## Introduction

Wetlands such as floodplains, estuaries, and mangrove forests are valuable nurseries for a wide range of commercially, recreationally, and ecologically important aquatic species [1, 2]. The loss of juvenile 'nursery habitats' are especially detrimental to recruitment-dynamics in fish populations because these habitats disproportionately contribute to growth and survival of individuals that recruit into adult populations and fisheries [3]. Identifying, conserving, and restoring nursery habitats is a conservation management priority because their rapid rate of loss impacts the production of freshwater and diadromous fishes important for food production globally [4, 5]. Despite the perceived value of wetland habitats, studies that directly quantify whether they are disproportionately important to recruitment success for key fish species are limited [3, 6, 7]. This is largely due to the challenges in tracking habitat use by fish across multiple life-stages and heterogeneous habitats at landscape scales.

Stable isotope analysis (SIA) of carbon ($\delta^{13}$C), nitrogen ($\delta^{15}$N), and sulfur ($\delta^{34}$S) has emerged as a tool to track fish movement, diet, and habitat-use across aquatic landscapes that differ in their food webs, particularly across freshwater, estuarine, and marine habitats [8–15]. Traditional SIA applications typically require sampling multiple tissues (blood, scales, liver, and muscle) from the same individual with varying tissue turnover rates (days, weeks, months) in order to track changes in $\delta^{13}$C and $\delta^{15}$N over time and infer fish movement and/or diet shifts [16, 17]. Because the food web communities on which juvenile fish forage can differ predictably between habitats, SIA in tissues can provide time resolved movements through diverse habitats and life stages [9, 11, 12, 17]. For example, $\delta^{13}$C and $\delta^{34}$S in liver and muscle tissue have been used to quantify the time juvenile salmon spent in estuarine habitats [12]. However, studies that aim to use SIA in fish tissues to quantify the role of different habitats to survival into subsequent life stages are constrained by the tissue with the longest turn-over rate due to depuration over time (months; [17]).

Isotope and elemental incorporation into tissues such as otoliths in fishes can provide a permanent chemical archive of environmental and diet conditions throughout the life of a fish [18, 19]. Otoliths consist of calcium carbonate that is precipitated daily on a non-collagenous protein matrix forming incremental bands that preserve a record of age, growth, and environmental conditions [18, 19]. Measurements of stable isotopes bound to proteins through dietary pathways in otoliths are challenging due to the minimal amount of organic material in this calcium carbonate structure [19]. For example, sulfur is incorporated into the protein matrix [20] and is considered to primarily represent dietary input at the time of increment formation [21–23]. However, inorganic sulfur sourced from the water may also contribute [24]. Studies that have used SIA in otoliths for diet reconstructions often resort to whole otolith assays or coarse micro-drilling, thereby foregoing high-resolution chronologies even if differences exist over an individual's life history [10, 11, 25, 26]. In-situ measurements of $^{34}$S in otoliths at high spatial resolution (30μm spot sizes) allows for the detection in dietary shifts at the scale of 6–10 days, although smaller spot sizes are possible providing even finer temporal resolution [21, 23].This method has been applied to distinguish hatchery and wild salmon [21, 23, 27] as well as to determine anadromy in Sockeye/Kokanee Salmon in British Columbia due to differences between fresh water and marine diets, yet these studies are often limited to small sample sizes due to analytical and time constraints [22].

In the Pacific Northwest, floodplains are thought to be critical rearing habitats for many salmon species due to high primary and secondary food production, accelerated growth, and reduced predation [28–32]. Therefore, identifying the ecological role of floodplains for salmon and identifying whether they are functioning as nurseries is critical for understanding salmon ecology and improving management. Testing the nursery role of floodplains requires tracking the fate of individuals that used the floodplain compared with the fate of individuals that used alternative habitats as juveniles. This evaluation could be achieved through development and application of a natural floodplain habitat marker.

An ideal floodplain rearing marker would 1) be distinguishable from other possible salmon rearing habitats across the landscape 2) remain stable within season and among years, 3) become integrated into tissues at known equilibration rates, and 4) be permanently stored in archival tissues such as otoliths. Identifying such a marker is challenging because floodplains are ephemeral bodies of water that are dynamic in space and time. Water and its geochemical composition (e.g. Sr, Ca, Mn, Ba, $^{87}Sr/^{86}Sr$) traditionally measured in biominerals such as otoliths are unlikely to differ between main stem rivers and floodplains that share common source waters [33]. Hydrogen ($\delta^2H$) isotopes show promise given their variability across the freshwater landscape, with a linear relationship between presence in water and muscle tissue, and water and otoliths [10]. Yet, high temporal and spatial variability from evaporation and/or mixing of multiple water sources, makes $\delta^2H$ a temporally variable and unreliable marker to reconstruct floodplain occupancy [34].

High residence time of water on wetland and floodplains can support highly productive benthic food webs fueled by decomposition of plants and organic matter through anaerobic metabolic pathways [35, 36]. Under these conditions, methane-oxidizing bacteria preferentially take up carbon's lighter isotope providing a lower $\delta^{13}C$ value to the zooplankton and macroinvertebrates that consume them [37]. Similarly, sulfate is reduced to hydrogen sulfide ($H_2S$) by sulfate-reducing bacteria, resulting in low $\delta^{34}S$ values in sulfur-bearing compounds at the base of the food web [38, 39]. In floodplains in the Amazon and in North America, $\delta^{13}C$ has been used to trace energy flow between terrestrial and aquatic carbon sources, while $\delta^{15}N$ has been frequently used to trace relative trophic positions of food web components [11, 14, 37, 40–42]. In contrast, there is a natural salinity gradient with $\delta^{34}S$ becoming increasingly enriched as salinity increases which has been useful for habitat differentiation and fish migration [12, 43, 44]

Here, we explore $\delta^{13}C$, $\delta^{15}N$, and $\delta^{34}S$ as potential floodplain markers using Chinook Salmon (*Oncorhynchus tshawytscha*) in California's Central Valley (CCV) as a model system (Fig 1). Floodplains in the CCV are large landscape features, making this an idea model system to characterize biogeochemical processes and isotopic markers relevant to other systems. Floodplains are a primary feature of the landscape in the Sacramento-San Joaquin watershed; they are available to salmon during years with above average precipitation [31, 45, 46]. Because of the large area of rearing habitat that becomes inundated (Yolo Bypass = 23,876 hectares; Sutter Bypass = 6,300 hectares [31, 47]), these seasonal off-channel habitats provide accelerated growth for juvenile salmon and may therefore also serve a nursery function important to salmon population dynamics [31, 48]. Developing a floodplain marker to track floodplain use should help to determine the extent to which juveniles that rear on the floodplain from the distinct runs of salmon that contribute to the fishery, and return to spawn.

In order to evaluate whether stable isotopes could be used as robust floodplain makers, we conducted a large-scale field experiment to test 1) taxonomic and isotopic composition ($\delta^{13}C$, $\delta^{15}N$, $\delta^{34}S$) of salmon diets between salmon that fed on local prey in the floodplain vs in the river, 2) assimilation rates of $\delta^{13}C$ and $\delta^{34}S$ into muscle tissues under fluctuating natural environmental and hydrologic conditions, 3) temporal stability of the stable isotope values between

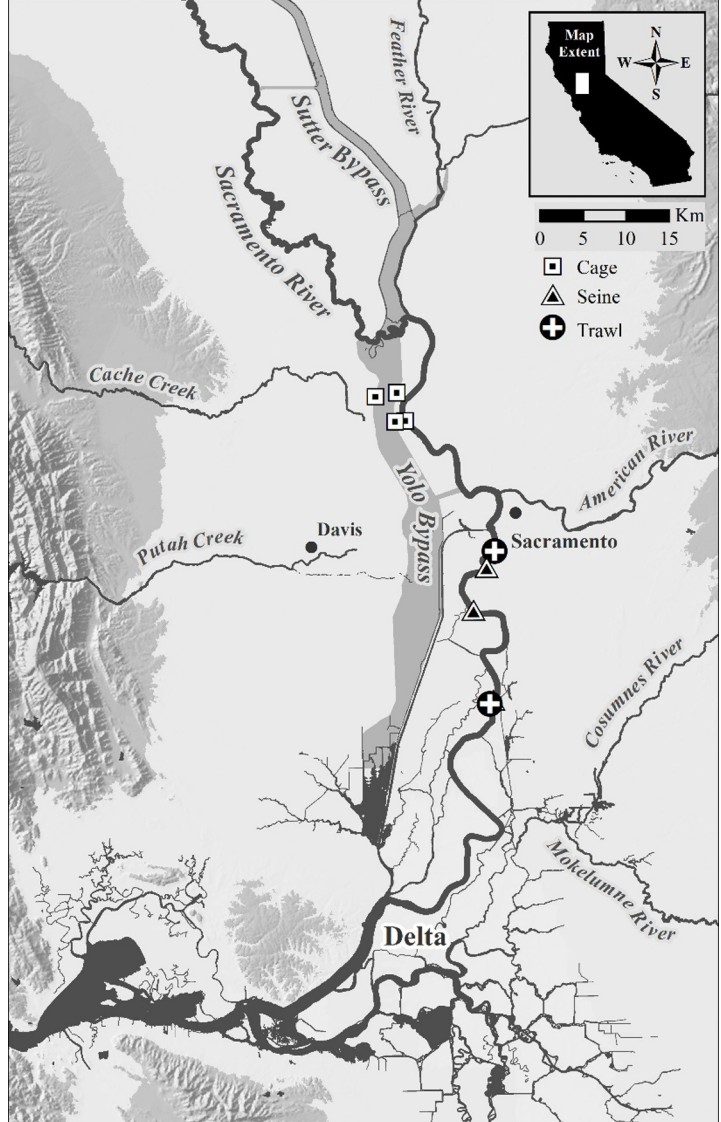

**Fig 1. Map of the Sacramento River and upper San Francisco Estuary, showing the Yolo Bypass and sample locations.**

habitats among years of varying hydrological conditions (1999–2017), and 4) $\delta^{34}$S isotopic values at multiple time scales—invertebrate prey items (days), muscle tissues (months), and weekly diet chronology permanently archived in otoliths (lifetime) from juveniles reared in the floodplain and riverine habitats.

## Methods

### Study system

The CCV supports four genetically distinct runs of Chinook Salmon at their southernmost native distribution [49]. In the CCV, more than 95% of the historic freshwater wetland habitats once used by juvenile salmonids have been lost to human development [49, 50]. Studies have demonstrated that size and timing of ocean entry [51] and freshwater growth rates [27] are

important factors influencing survival of CCV salmon, highlighting the importance of providing high quality freshwater habitats for salmon.

California's Mediterranean climate is highly variable with periods of both extreme droughts and floods [52, 53]. The Sacramento River (Fig 1) is California's largest river and has been heavily modified for flood control, water conveyance, and human use [54]. The lower 245 km of the Sacramento River are channelized and leveed, effectively reducing the amount of natural floodplain [47]. This disconnection of the floodplain decreased phytoplankton production and diminished access to detrital floodplain food webs, decreasing the overall productive capacity of the river food web and contributing to the decline in many fish species [55].

Depending on hydrologic conditions within the Central Valley, there are two main rearing pathways (floodplain and river) for juvenile Chinook Salmon prior to entering the Delta as they make their way to the San Francisco Estuary and then to the Pacific Ocean. During high flow events in the Sacramento River, juvenile salmon can gain access to remnant floodplains such as the Yolo Bypass (Fig 1) or remain in the mainstem Sacramento River. In the Bypass, water spills over Fremont weir, flooding up to 23,876 hectares of ephemeral floodplain and tidal habitat [31]. During low-water conditions fish are mostly confined to the leveed Sacramento River channel- although they can still access the lower tidal reaches of the Yolo Bypass and the upstream Sutter Bypass [32, 48].

In the Yolo Bypass, hydrologic connection of river and floodplain during floods provide fish access to floodplain habitats in about 70% of years [31, 45, 46]. Decomposition of abundant organic matter and in-situ phytoplankton production in the Yolo Bypass provides an important base for an aquatic food web which supports a large biomass of zooplankton and other invertebrates [29, 56]. These high densities of important prey items promote higher growth rates in juvenile Chinook Salmon rearing on the floodplain compared to those confined to the adjacent riverine habitat [28, 30, 31, 36]. In wet years with prolonged connectivity between floodplain and river and substantial inundation of floodplain habitats, rearing on the Yolo Bypass contributes to increased variation in outmigration timing, growth rates and phenotypic diversity, enhancing sources of resilience in Chinook Salmon populations [32, 57]. Taken together, these characteristics identify the Yolo Bypass as an important rearing habitat for young-of-the-year Central Valley Chinook Salmon. For this reason, the floodplain has become a major focus of habitat restoration activities to support endangered runs of salmon. Overall, survival of juveniles that are moving down through the main channel of the Sacramento River is relatively low [58]. Although coded wire tag and acoustic fish tagging studies have provided evidence for the elevated survival of larger juvenile fish (fork length > 80mm) through Yolo Bypass, the contribution of these fish to the population remains difficult to quantify [48]. Further, fish that are too small to acoustically tag are typically the life stage most likely to benefit from floodplain rearing [27, 31].

## Sample collection

For this study, juvenile Chinook Salmon were reared on a managed floodplain as well as caged in the adjacent river habitat across multiple years. Concurrently, fish were captured in the Sacramento River. More specifically, during the winter wet seasons of 2014–2016, several thousand (5,866–42,600; Supplemental A in S1 File, [36]) juvenile Chinook Salmon from the Feather River Hatchery were reared on nine 0.81ha replicated, shallow inundated floodplain habitats constructed on agricultural fields within the Yolo Bypass (Fig 1) [28]. The fields were farmed during the summer growing season. Juvenile salmon were reared on the winter-flooded rice fields for 4–6 weeks, to approximate a typical amount of time juvenile salmon might spend on the naturally inundated Yolo Bypass floodplain during flood events before

migrating downstream [31, 48]. To test whether the floodplain food web was taxonomically and isotopically distinct, stomach contents and muscle tissue were sampled weekly from 5–10 fish. Differences in stomach content composition and associated stable isotope values were expected based on published data [29, 36, 59], and on initial sulfur isotope mapping of the Sacramento River and upper San Francisco Estuary watershed (Supplemental B and S1 Fig in S1 File). Fish were collected with a 15m beach seine or captured in outmigration traps constructed at the outflow of the experimental habitats. Each outmigration trap was 150cm x 91cm x 91cm and made of 3mm plastic mesh and affixed to the field's outlet structure. Fish were collected in the outmigration trap in 2016 only (Supplemental A in S1 File). After sample collection, fish were transported on ice back to the laboratory and frozen until dissected.

Actively out-migrating juvenile Chinook Salmon were also captured during 2012–2017 at multiple sites along the Sacramento River near the city of Sacramento, parallel to the Yolo Bypass as part of long-term USFWS beach seine sampling program [31]. Approximately 30–60 juveniles were collected per year using a 15m beach seine, midwater trawl, and/or Kodiak trawl [60].

Because actively out-migrating Chinook Salmon can experience a variety of habitats before being captured, fish were placed within enclosures during 2016 and 2017 in both the Yolo Bypass floodplain and the Sacramento River to ensure diets reflected habitats of interest. During both years, 10 fish were placed into three $1.2 \times 1.2 \times 0.6$m enclosures (Fig 1 and Table 1 and Supplemental A in S1 File). The enclosure frames were constructed from 19mm polyvinyl chloride (PVC) pipe with 6.3mm extruded plastic netting fitted around the frame. Cages were cleaned weekly to ensure prey availability was comparable to free swimming fish. Due to the difficulty of managing enclosures during high flow events within the river, fish reared in these cages for approximately 4 weeks. At the end of the study, 5 fish from each habitat in each year were euthanized and transported on ice to the lab, where they were frozen until thawed for dissection (Table 1, Fig 1 and Supplemental A in S1 File).

During the high water years of 1999 and 2017 when juvenile Chinook Salmon could access Yolo Bypass in floodwaters from the Sacramento River, fish were captured using seines in shallow, peripheral habitats throughout Yolo Bypass with methods established by Sommer et al. [31]. In 2016, Fremont Weir overtopped briefly allowing for opportunistic sampling of wild fish naturally recruited to the Yolo Bypass floodplain. These wild fish moved into the experimental floodplains volitionally during the flood and were captured during outmigration (S2 Fig in S1 File).

Sampling coincided with one of California's most severe droughts (2014–2015), an average year (2016), and with the wettest year (2017) on record (North Sierra Precipitation: 8 station

**Table 1. Chronology of data collection with the tissue types and isotope systems analyzed.**

|  | 1999 | 2012 | 2013 | 2014 | 2015 | 2016 | 2017 |
|---|---|---|---|---|---|---|---|
| **Pilot Data** | Stomach$^{\delta 34S}$ |  |  |  |  |  |  |
|  | Muscle$^{\delta 34S}$ |  |  |  |  |  |  |
| **Opportunistic** |  |  |  |  |  | Stomach$^{\delta 13C,\ \delta 15N,\ \delta 34S}$ | Stomach$^{\delta 13C,\ \delta 15N,\ \delta 34S}$ |
|  |  |  |  |  |  | Muscle$^{\delta 13C,\ \delta 15N,\ \delta 34S}$ | Muscle$^{\delta 13C,\ \delta 15N,\ \delta 34S}$ |
| **Experimental fields** |  |  |  | Stomach$^{\delta 13C,\ \delta 15N,\ \delta 34S}$ | Stomach$^{\delta 13C,\ \delta 15N,\ \delta 34S}$ | Stomach$^{\delta 13C,\ \delta 15N,\ \delta 34S}$ | Stomach$^{\delta 13C,\ \delta 15N,\ \delta 34S}$ |
|  |  |  |  | Muscle$^{\delta 13C,\ \delta 15N,\ \delta 34S}$ | Muscle$^{\delta 13C,\ \delta 15N,\ \delta 34S}$ | Muscle$^{\delta 13C,\ \delta 15N,\ \delta 34S}$ | Muscle$^{\delta 13C,\ \delta 15N,\ \delta 34S}$ |
| **Monitoring Data** |  | Stomach$^{\delta 34S}$ | Stomach$^{\delta 34S}$ | Stomach$^{\delta 13C,\ \delta 15N,\ \delta 34S}$ | Stomach$^{\delta 13C,\ \delta 15N,\ \delta 34S}$ | Stomach$^{\delta 13C,\ \delta 15N,\ \delta 34S}$ | Stomach$^{\delta 13C,\ \delta 15N,\ \delta 34S}$ |
|  |  | Muscle$^{\delta 34S}$ | Muscle$^{\delta 34S}$ | Muscle$^{\delta 13C,\ \delta 15N,\ \delta 34S}$ | Muscle$^{\delta 13C,\ \delta 15N,\ \delta 34S}$ | Muscle$^{\delta 13C,\ \delta 15N,\ \delta 34S}$ | Muscle$^{\delta 13C,\ \delta 15N,\ \delta 34S}$ |
| **Caged Data** |  |  |  |  |  | Stomach$^{\delta 13C,\ \delta 15N,\ \delta 34S}$ | Stomach$^{\delta 13C,\ \delta 15N,\ \delta 34S}$ |
|  |  |  |  |  |  | Muscle$^{\delta 13C,\ \delta 15N,\ \delta 34S}$ | Muscle$^{\delta 13C,\ \delta 15N,\ \delta 34S}$ |
|  |  |  |  |  |  | Otolith$^{\delta 34S}$ |  |

index; Department of Water Resources, 2018; S2 Fig in S1 File) [61]. Hydrologic variability of this magnitude provided an opportunity to explore the isotopic variation of the floodplain and river across variable environmental conditions including flood (1999, 2017; wet water years), average (2016; below normal water year) and drought (2012–2015; dry and critical water years) (S2 Fig in S1 File).

Fish were euthanized in the field by directed concussive impact to cranial foci as per protocols approved by University of California Davis IACUC #'s 20979 and 17137.

## Taxonomic and isotopic composition ($\delta^{13}$C, $\delta^{15}$N, $\delta^{34}$S) of salmon diets

Invertebrate prey items were extracted from all juvenile salmon stomachs (N = 365) and identified to the lowest taxonomic group using aquatic invertebrate identification keys [62, 63]. Stomach contents were analyzed at the UC Davis Stable Isotope Facility, which provides combined $\delta^{13}$C, $\delta^{15}$N, $\delta^{34}$S isotope values from a single sample. Fish with empty or nearly empty guts were eliminated from isotopic analysis. Stomach contents from fish whose stomachs were extremely full were homogenized and subsampled to meet a 2-5mg dried weight for combined $\delta^{13}$C, $\delta^{15}$N, $\delta^{34}$S isotope analysis. Stomach contents were placed into 8 x 5mm tin capsules (Elemental Microanalysis pressed tin capsules) or if the invertebrate prey items were larger, then 10 x 8mm tin capsules. Each capsule was then placed in a 96-well plate and dried for 48 hours in a drying oven with temperatures not exceeding 55˚C. Tins were weighed, crimped, and folded into small discs, and then placed back into the 96-well plate for isotopic analysis. For $\delta^{13}$C and $\delta^{15}$N analyses, samples were analyzed using an Elemental Analyzer–Isotope Ratio Mass Spectrometer (EA-IRMS). For $\delta^{34}$S in the solid samples, we used an EA-IRMS specifically designed for $\delta^{34}$S measurements in animal tissue.

Differences in diet composition between river and floodplain habitats were tested with a Pearson's chi-squared test applied to the estimated taxonomic classifications for all fish reared or captured in the floodplain and river across all years. To test whether each habitat's dietary isotopic value could be used to classify individuals into the correct habitat of origin (river or floodplain), a linear discriminant analysis (LDA) with jackknife cross validation was run separately for $\delta^{13}$C and $\delta^{34}$S, and for $\delta^{13}$C + $\delta^{34}$S combined using the MASS package [64] in R. Only stomach content data were included in the LDA because we were interested in testing the differences at the base of the food webs present in the different habitats. Muscle isotope data were excluded from the LDA because they represent a time integrated value that is not necessarily in equilibrium with the habitat at the location of sample collection. Only individuals with both $\delta^{13}$C and $\delta^{34}$S values (N = 315) were used in the LDA. Prior to analysis the data were inspected using qq plots for univariate normality and boxplots for homogeneity of variance between groups.

Mixed effects models were used to understand variation in $\delta^{13}$C and $\delta^{34}$S in invertebrate prey items from juvenile Chinook Salmon in different environmental conditions, and from different locations within the floodplain. Specifically, we modeled the isotope values in response to rearing habitat (river or floodplain) and then used environmental conditions (flood, average, and drought) as a random effect to account for different annual-scale hydrologic conditions and increased sampling efforts as more water inundates the bypass. Since increased sampling efforts allowed us to capture fish naturally recruited to the floodplain in addition to those reared in the experimental ponds, the location (subsite) within the floodplain where a fish was captured was included as a random effect in the model to account for volitional fish. These models were run for $\delta^{13}$C and $\delta^{34}$S separately. Models were compared using a likelihood ratio test to evaluate which random effect had the most influence on $\delta^{13}$C and $\delta^{34}$S values. Caged fish were not included as a random effect because cages in the river were only

used for two years of this multi-year study, and consequently there were not enough samples to include in the model. Only years with direct river to floodplain comparisons were used in the mixed effect model (2014–2017). The lme4 package was used to analyze the isotopic invertebrate prey item data as a two-way repeated measures analysis of variance (S2 Table in S1 File) [65]. Data were scaled and centered prior to analysis. An alpha value of $P < 0.05$ was used to test for significance. All statistical analyses were performed using R software version 4.0.2 (R Core Team, 2020).

Juvenile Chinook Salmon are likely to feed at the same trophic position in the floodplain and the river. However, to test this, the mean $\delta^{15}N$ values of invertebrate prey items found in the stomach contents between floodplain and river fish across all years were compared with a t-test.

## Diet-shit experiment and muscle isotope values ($\delta^{13}C$, $\delta^{15}N$, $\delta^{34}S$)

To assess the relationship between the duration of time a fish reared on the floodplain and isotopic values in fish tissues, we conducted a diet-shift field experiment. Hatchery fish that had been fed a marine based diet with enriched $\delta^{34}S$ values (+14.1‰ to +16.6‰; [21]) were transferred to the experimental fields within the floodplain and with access to natural prey items. The tissue assimilation experiments occurred during the years 2014–2016, for 6 weeks when weekly sampling was able to take place on the experimental fields. In 2015, the experiment was suspended early, after 4 weeks, due to increasing temperatures that were potentially lethal to the juvenile Chinook Salmon.

For stable isotope analysis a small amount of muscle tissue (~3mg dry weight) was removed from just below the dorsal fin of each fish. Dried muscle tissue from the same individuals were split with 1mg analyzed for $\delta^{13}C$ and $\delta^{15}N$ and the other 2-5mg analyzed separately for $\delta^{34}S$. Samples were analyzed at the UC Davis Stable Isotope Facility using the same analytical equipment and procedures used for stomach contents.

## In situ Otolith sulfur isotope ($\delta^{34}S$) analyses

To determine if $\delta^{34}S$ found in invertebrate prey items was permanently archived in otoliths, we subsampled five hatchery origin fish from the 2016 tissue assimilation study. Of these five, three were fish that reared on the floodplain for the full duration of the study in 2016 (39 days) and two were fish that had reared within the cages in the Sacramento River in 2016 (22 days). A limited number of representative samples (N = 5) were selected due to the scope of the current project to serve as a proof of concept of the utility of sulfur measurement in otoliths for future applications. Juvenile Chinook Salmon sagittal otoliths were extracted, cleaned with ultra-pure water, and stored dry before being mounted in Epoxicure (Buehler Scientific) epoxy resin. Otoliths were polished on both sides to reveal internal structure so that the daily growth bands after the exogenous feeding check were visible. Then the otoliths were digitized with a 12-megapixel digital camera attached to an Olympus CH30 compound microscope with a 20X objective, using AM Scope (MU1000). Daily otolith increments were enumerated and the increment width and radial distance (μm) from the core to each daily ring was measured using Image Pro Premier (Media Cybernetics Inc., Rockville, MD). Aging transects followed the dorsal plane of the otolith at about 90 degrees from the anterior-posterior axis. The number of days' salmon reared in their enclosures in the floodplains and rivers was known and therefore, the day/location on the otolith corresponding to the experimental shift in diet was backcalculated and identified. Prior to analysis, samples were mounted in epoxy, degreased with petroleum ether, and cleaned with RBS detergent and Millipore H2O. The mounts were then dried in a 60˚C vacuum oven for at least 24h and coated with high purity Au.

In situ sulfur isotope values were measured using the Sensitive High Resolution Ion Micro Probe (SHRIMP II) at the Research School of Earth Sciences, The Australian National University (ANU), Canberra, Australia. The SHRIMP II was operated in negative ion mode and followed established protocols for oxygen isotope analyses [66–68]. In brief, a 15kV, ~ 3nA $Cs^+$ primary ion beam was focused to a spot ~ 30μm diameter on the Au-coated target surface, resulting in a sputtered pit ~ 2μm deep representing approximately one-week of otolith deposition and fish growth. Negative $S^-$ secondary ions were measured in multiple collector mode using Faraday cup detectors and custom-built iFlex electrometers operated in resistor (current) mode for $^{32}S$ ($10^{12}$ Ω) and in charge mode for $^{34}S$. Each analysis took seven minutes and consisted of 120s pre-sputtering during which the primary beam rastered an area slightly larger than the analytical pit to remove any surface contamination. The secondary ion was then stabilized, and electrometer baselines were measured, followed by ~ 100s of automated centering of the secondary ion beam steering to maximize the secondary ion count rates, and 400s of data collection. Data acquisition consisted of twenty measurements (20 s each) for each spot, resulting in an internal precision of < 2‰ (95% CI) for $\delta^{34}S$. Data reduction was performed offline using the POXI software. Typical beam currents on $^{32}S^-$ and $^{34}S^-$ were 1.6x10$^{-13}$A and 6.4x10$^{-15}$A, respectively. Backgrounds were typically 1x10$^{-15}$A ($10^{12}$ Ω resistor) and 1x10-17 A (charge mode), respectively. The $^{34}S/^{32}S$ ratios are reported as $\delta^{34}S$ values in parts per mil relative to the Vienna-Canyon Diablo Troilite (VCDT, $^{34}S/^{32}S$ = 0.044163) standard [69] as shown in Eq 1.

$$\delta^{34}S = \left( \frac{{}^{34}S/{}^{32}S\ sample}{{}^{34}S/{}^{32}S\ VCDT} \right) - 1 \qquad (1)$$

Since the absolute sulfur isotopic composition of these otoliths has not been established and no standard material for $\delta^{34}S$ in otoliths was available we accounted for instrumental mass fractionation by repeated measurements of the otolith core and assuming that this is representing the marine value of $\delta^{34}S$ = 18‰, Eq 2.

$$\delta^{34}S_{sample-cor} = \delta^{34}S_{sample-raw} - (\delta^{34}S_{ref-raw} - \delta^{34}S_{ref-est}) \qquad (2)$$

Here, $\delta^{34}S_{sample-cor}$ is the corrected value for a sample spot, $\delta^{34}S_{sample-raw}$, the uncorrected measured value, and $\delta^{34}S_{ref-raw}$ and $\delta^{34}S_{ref-est}$, are the uncorrected measured value for the core and the estimated value (18‰), respectively. While there are significant uncertainties that can be introduced through this internal standardization method, the relative differences among the $\delta^{34}S$ values of the analyzed otoliths are not affected. Errors on individual spot measurements are given at the 95% confidence interval level.

Analytical profiles started at the ventral edge and then traversed through the core to the dorsal edge with a spacing of ~ 10μm in-between spots. Spots were placed parallel to aging transect on the dorsal lobe to allow for the conversion from analytical distance to age in days. After SIMS analyses the locations of the spots were visually confirmed using a microscope (LEICA DM6000_M) to identify the spots that corresponded to time in the floodplain or river and ensure that the analysis spots did not overlap with any contaminants (e.g., epoxy), irregular surface morphology (cracks, breaks), or vaterite (calcium-carbonate polymorph), as these could influence the $\delta^{34}S$ value.

## Results

To evaluate our ability to detect differences in river and floodplain habitat use of juvenile Chinook Salmon, we first report the results of the taxonomic composition of stomach contents,

evaluate the ability of stable isotope analysis to classify habitat-use using an LDA approach, and then test the temporal stability among different hydrologic conditions using mixed effects models. We then present muscle tissue isotopic depuration over time to quantify the number of days required to reach equilibrium with source diets. Lastly, we display a time series of isotopic values in otoliths to investigate if these biominerals can serve as a permanent record of dietary shifts useful to track juvenile habitat use.

## Taxonomic and isotopic composition of salmon diets

**Taxonomic differences in stomach contents.** The taxonomic composition of stomach contents of fish reared on the floodplain compared to those reared in the Sacramento River showed consistent differences in all years (Fig 2). Floodplain fish fed primarily on Cladoceran-dominated zooplankton while fish captured in the river fed on a diversity of macroinvertebrates and zooplankton (Fig 2). Diet composition was significantly associated with habitat location (Chi Squared, $X^2$ = 12241, df = 6, P < 0.0001).

**Isotopic ($\delta^{13}$C, $\delta^{15}$N, $\delta^{34}$S) differences in stomach contents.** The $\delta^{13}$C and $\delta^{34}$S values of the invertebrate prey items from stomachs of juvenile Chinook Salmon showed a clear separation between floodplain and river habitats (Table 2 and Fig 3 and S1 Table in S1 File). The combined $\delta^{13}$C and $\delta^{34}$S LDA, based on stomach contents where both measurements were available (N = 315), correctly re-assign each observation back to its habitat-of-origin with high accuracy (94% accuracy, CI 0.91–0.97, kappa = 0.88). LDA on stomach contents performed using only $\delta^{34}$S still achieved comparably high classification success (92% accuracy, CI 89–95%, kappa = 84%), while only using $\delta^{13}$C resulted in lower re-classification success (89% accuracy, CI 85–92%, kappa = 76%).

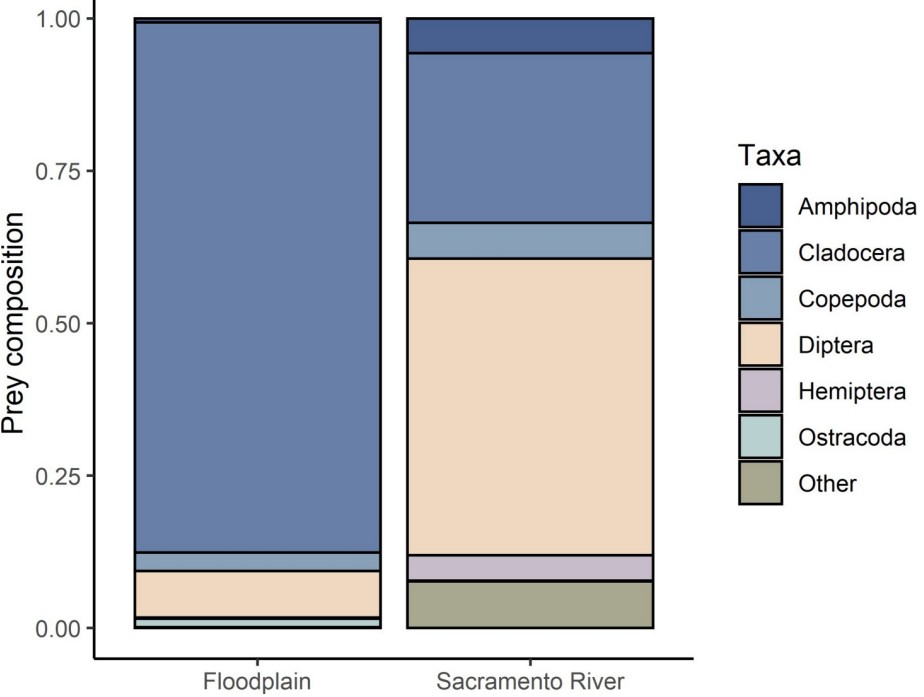

**Fig 2. Relative diet composition from fish captured in the floodplain and adjacent river by taxa, all years averaged together (N = 365).**

**Table 2. Fish stomach content stable isotope data ($\delta^{13}$C, $\delta^{15}$N, $\delta^{34}$S) for different hydrologic conditions from river and floodplain habitats.** Enclosed sites include fish that were reared on the experimental fields or in cages.

| Habitat | Site | Year type | $\delta^{13}$C mean | $\delta^{13}$C SD | $\delta^{13}$C n | $\delta^{15}$N mean | $\delta^{15}$N SD | $\delta^{15}$N n | $\delta^{34}$S mean | $\delta^{34}$S SD | $\delta^{34}$S n |
|---|---|---|---|---|---|---|---|---|---|---|---|
| **River** | Sacramento River | average | -27.91 | 1.62 | 54 | 7.36 | 1.1 | 54 | 2.77 | 2.52 | 58 |
| | Sacramento River | drought | -28.27 | 2.16 | 75 | 7.16 | 1.72 | 75 | 1.91 | 2.43 | 84 |
| | Sacramento River | flood | -28.83 | 1.35 | 50 | 7.06 | 1.36 | 50 | 2.28 | 1.64 | 50 |
| | Sacramento River Enclosed | average | -28.03 | 1.8 | 5 | 7.53 | 0.88 | 5 | 1.84 | 0.99 | 5 |
| | Sacramento River Enclosed | drought | -27.99 | | 1 | 10.18 | | 1 | -0.8 | | 1 |
| | Sacramento River Enclosed | flood | -30.65 | 0.5 | 5 | 8.54 | 1.03 | 5 | 2.03 | 0.41 | 5 |
| **Floodplain** | Yolo Bypass | average | -30.76 | 0.74 | 13 | 7.74 | 1.28 | 13 | -1.49 | 1.45 | 13 |
| | Yolo Bypass | flood | -32.17 | 3.17 | 27 | 7.64 | 0.82 | 27 | -1.86 | 1.14 | 28 |
| | Yolo Bypass Enclosed | average | -34.49 | 2.62 | 32 | 7.68 | 1.41 | 32 | -4.11 | 1.3 | 32 |
| | Yolo Bypass Enclosed | drought | -33.53 | 1.77 | 55 | 6.52 | 0.99 | 55 | -4.52 | 2.59 | 55 |
| | Yolo Bypass Enclosed | flood | -28.41 | 0.5 | 5 | 7.45 | 0.14 | 5 | -1.96 | 0.46 | 5 |

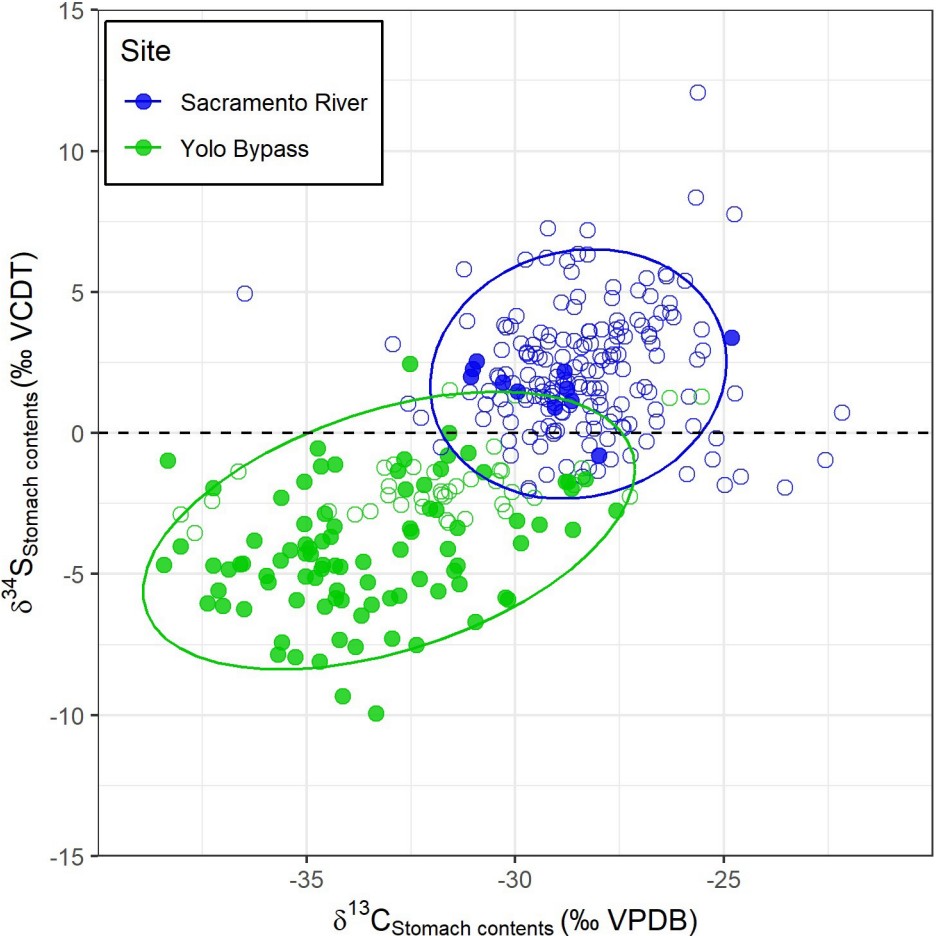

**Fig 3. $\delta^{13}$C and $\delta^{34}$S values from stomach contents from the same fish analyzed for diet composition.** Fish captured/reared from the floodplain are shown in green and from the river in blue. Open circles represent fish captured in the river or naturally recruited onto the floodplain while closed circles represent fish from field and caged experiments. Ellipses represent 95% confidence intervals.

The mixed effects models (S2 Table in S1 File), showed that there is significant spatial variation in both $\delta^{13}$C and $\delta^{34}$S ($\delta^{13}$C ANOVA, $X^2$ = 70.41, P < 0.001; $\delta^{34}$S ANOVA, $X^2$ = 13.93, P < 0.001) within the Yolo Bypass (Fig 4), but the isotope values generally remained lower compared to the river values. We did not see significant isotopic variation in $\delta^{34}$S among different hydrologic conditions from fish reared or captured on the Yolo Bypass ($\delta^{34}$S ANOVA, $X^2$ = 2.58, P = 0.11) (Fig 4). However, there was isotopic variation in $\delta^{13}$C values from the invertebrate prey items depending on the hydrologic conditions in the Yolo Bypass ($\delta^{13}$C ANOVA, $X^2$ = 3.97, P = 0.05).

Comparisons of $\delta^{15}$N values from the invertebrate prey items were similar (T-test, t = -0.45, df = 309.2, P = 0.66) between the two habitats (Yolo Bypass: $\delta^{15}$N = 7.19±1.22‰, Sacramento River: $\delta^{15}$N = 7.25±1.46‰, mean±1SD; S1 Table and S3 Fig in S1 File). There was minimal trophic difference, and thus no significant difference in $\delta^{15}$N values, between salmon feeding predominantly on zooplankton on the floodplain and those captured in the river feeding on a variety of zooplankton and macroinvertebrates.

## Isotopic equilibrium between stomach contents and muscle tissue

Diet-shift experiments were carried out in 2014–2016 to evaluate the rate at which muscle tissues reach equilibrium with stomach contents in the field. Muscle $\delta^{13}$C and $\delta^{34}$S values for hatchery-origin fish were elevated at the start of the experiments due to the marine-based protein diet on which juvenile salmon are fed at the hatchery (Fig 5) [70–72]. The $\delta^{13}$C and $\delta^{34}$S of

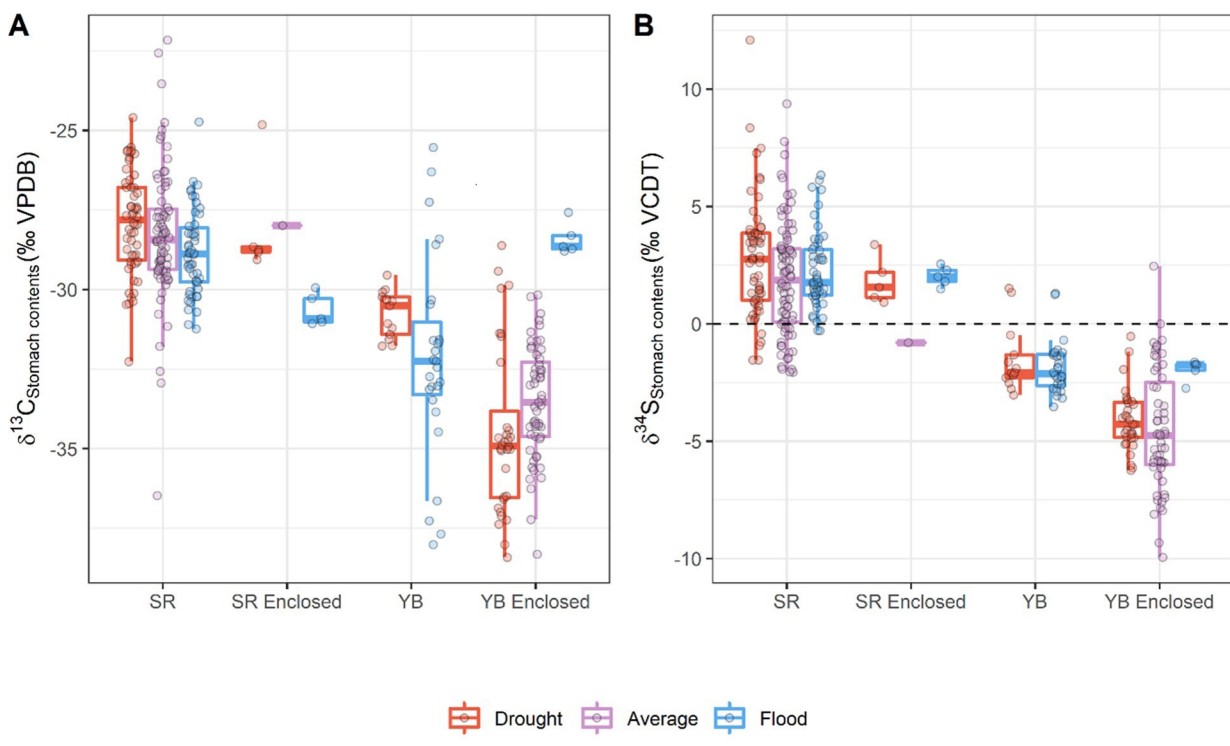

**Fig 4.** $\delta^{13}$C (A) and $\delta^{34}$S (B) values of stomach contents from river and floodplain habitats under different hydrologic conditions. $\delta^{34}$S values from the stomach contents of fish that reared on the floodplain were consistently lower throughout all years (2012–2015, drought; 2016, average; 1999 and 2017, flood). The $\delta^{13}$C values were also generally lower, but this trend varied with different hydrologic conditions. Box denotes the median and interquartile range, and whiskers denote 1.5 time the interquartile range.

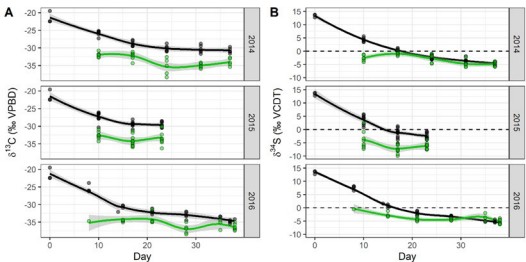

**Fig 5. Results from diet-shift experiment, including previously published data from the 2016 experiment [71].**
Each point represents a fish sampled over the duration of the study. The solid line is a loess smooth (span = 0.75) with a 95% confidence interval. Fish started with higher $\delta^{13}C$ (A) and $\delta^{34}S$ (B) values in their muscle tissue (black) because they were reared in a hatchery prior to the study. Over the course of the study, $\delta^{13}C$ became lower but displayed ~ 1‰ fractionation, as expected as prey items are integrated into tissue. Whereas their muscle tissues $\delta^{34}S$ values converged with the $\delta^{34}S$ values from the stomach contents (green).

the invertebrate prey items from fish sampled were consistent and stable over the course of the experiment, and muscle tissue isotope values reached equilibrium at 24 to 30 days (Fig 5) in 2014 and 2016. $\delta^{13}C$ displayed a ~ 1‰ fractionation as expected when prey items are integrated into tissue, while muscle $\delta^{34}S$ converged with the $\delta^{34}S$ from the stomach contents (Fig 5). The experiment was cut short in 2015, due to increasing water temperatures, which resulted in larger differences between stomach and muscle isotope values at the end of this experiment for most fish. The $\delta^{15}N$ values of stomach contents and muscle tissue showed a consistent offset of 3–4 ‰, as expected when prey items are integrated into tissue and no change over time in the trophic position of the diet occurs (S4 Fig in S1 File).

Wild-caught salmon sampled within the Yolo Bypass exhibited high variation in muscle tissue $\delta^{13}C$ and $\delta^{34}S$ values (Fig 6). The salmon that had been reared in either the experimental ponds or cages in the floodplain during this study had lower $\delta^{13}C$ and $\delta^{34}S$ values on average compared to the wild-caught salmon (Table 3 and Fig 6). As expected based on the lack of

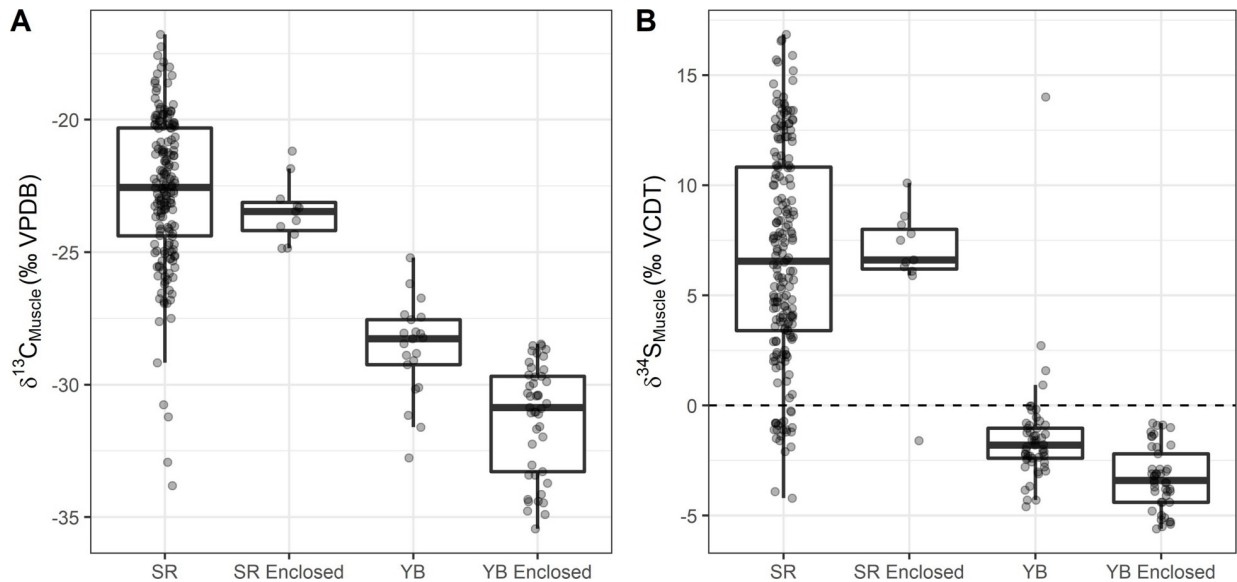

**Fig 6. $\delta^{13}C$ and $\delta^{34}S$ values in muscle tissues for wild-caught and enclosed (experimental fields and cages) fish in the river and floodplain habitats averaged for all years sampled.** Only muscle tissue data from enclosed fish reared more than 24 days of the time series were used to show the final isotope values assimilated into tissues.

**Table 3. Muscle stable isotope data for wild-caught and enclosed experimental fish from the river and floodplain habitats, averaged for all years sampled.**

| Habitat | Site | $\delta^{13}C$ mean | $\delta^{13}C$ SD | $\delta^{13}C$ n | $\delta^{15}N$ mean | $\delta^{15}N$ SD | $\delta^{15}N$ n | $\delta^{34}S$ mean | $\delta^{34}S$ SD | $\delta^{34}S$ n |
|---|---|---|---|---|---|---|---|---|---|---|
| River | Sacramento River | -22.67 | 2.82 | 180 | 12.04 | 2.11 | 180 | 6.77 | 4.87 | 196 |
| | Sacramento River Enclosed | -23.45 | 1.15 | 11 | 11.65 | 0.92 | 11 | 6.55 | 2.99 | 11 |
| Floodplain | Yolo Bypass | -28.64 | 1.79 | 21 | 10.31 | 1.16 | 21 | -1.45 | 2.56 | 53 |
| | Yolo Bypass Enclosed | -31.28 | 2.08 | 45 | 11.24 | 0.63 | 45 | -3.3 | 1.41 | 45 |

distinct floodplain $\delta^{15}N$ values in stomach content the $\delta^{15}N$ muscle tissue data did not provide a distinct signal to differentiate floodplain and river habitats (S5 Fig in S1 File).

The $\delta^{13}C$ and $\delta^{34}S$ values from muscle tissue of the fish captured throughout all years in the Sacramento River were generally higher than the floodplain samples but also highly variable (Table 3 and Fig 6). During 2016 and 2017 fish were cage-reared in the Sacramento River in addition to those captured in the river. The cage-reared fish in the river displayed less variation in their isotopic tissue values compared to fish collected from natural river habitats (Table 3 and Fig 6).

### Permanent records of diet-shifts

In situ $\delta^{34}S$ measurements in otoliths for five hatchery-origin chronicled weekly dietary differences in juveniles feeding in the floodplain relative to the river that was preserved over their lifetime (Fig 7 and S6 Fig in S1 File). A distinct change was observed in the $\delta^{34}S$ isotope profile, coinciding with the timing of movement from the hatchery to the river or floodplain. Average $\delta^{34}S$ values showed differences for the early juvenile period spent at the hatchery 14.72±1.04‰ (mean±SD), and 4.04±1.42‰ (mean±SD) for the entire river (22 days) or -3.27±0.08‰ (mean ±SD) for the entire floodplain period (39 days). The difference between river and floodplain $\delta^{34}S_{Otolith}$ is ~6–8‰, which is large enough to be detectable with in situ analytical methods at a weekly temporal resolution.

$\delta^{34}S$ measured at the otolith edge (30μm), representing the last 10–15 days, were compared to muscle and stomach content $\delta^{34}S$ values (Fig 8). $\delta^{34}S_{Otolith\ edge}$ values for floodplain fish were -5.60±0.16‰ and for river-reared fish $\delta^{34}S$ = 3.73±0.98‰. For fish reared on the floodplain $\delta^{34}S_{Otolith\ edge}$ values were consistent with $\delta^{34}S_{muscle}$ and $\delta^{34}S_{stomach}$ values of those specific fish, showing that these three tissues were in equilibrium. In contrast, $\delta^{34}S_{Otolith\ edge}$ values for river fish differed markedly, with $\delta^{34}S_{muscle}$ being higher and $\delta^{34}S_{stomach}$ being lower, indicating that both muscle and otolith were not in equilibrium with the local food source and/or that the local food source was variable.

## Discussion

Large river valleys are some of the most ecologically important landscapes on Earth [73], supporting high levels of biodiversity and important fisheries [74, 75]. Human alteration of these environments has created a biodiversity crisis and created extinction debts that conservation science is only now beginning to confront [76–78]. In this study, we present an example of how stable isotopes can be used to document seasonally inundated floodplain habitat use by juvenile Chinook Salmon. Our study demonstrated that $\delta^{34}S$ is an ideal floodplain marker that meets our established criteria of being distinct from alternative habitats, stable over time integrated into tissues at known equilibration rates, and stored in an archival tissue.

In 24–30 days, the muscle tissues were in equilibrium with the isotopic values found at the base of food web. This rapid equilibrium rate is likely due to the accelerated average growth rates (up to 1.28mm/day) seen in juvenile Chinook Salmon foraging on calorie-dense

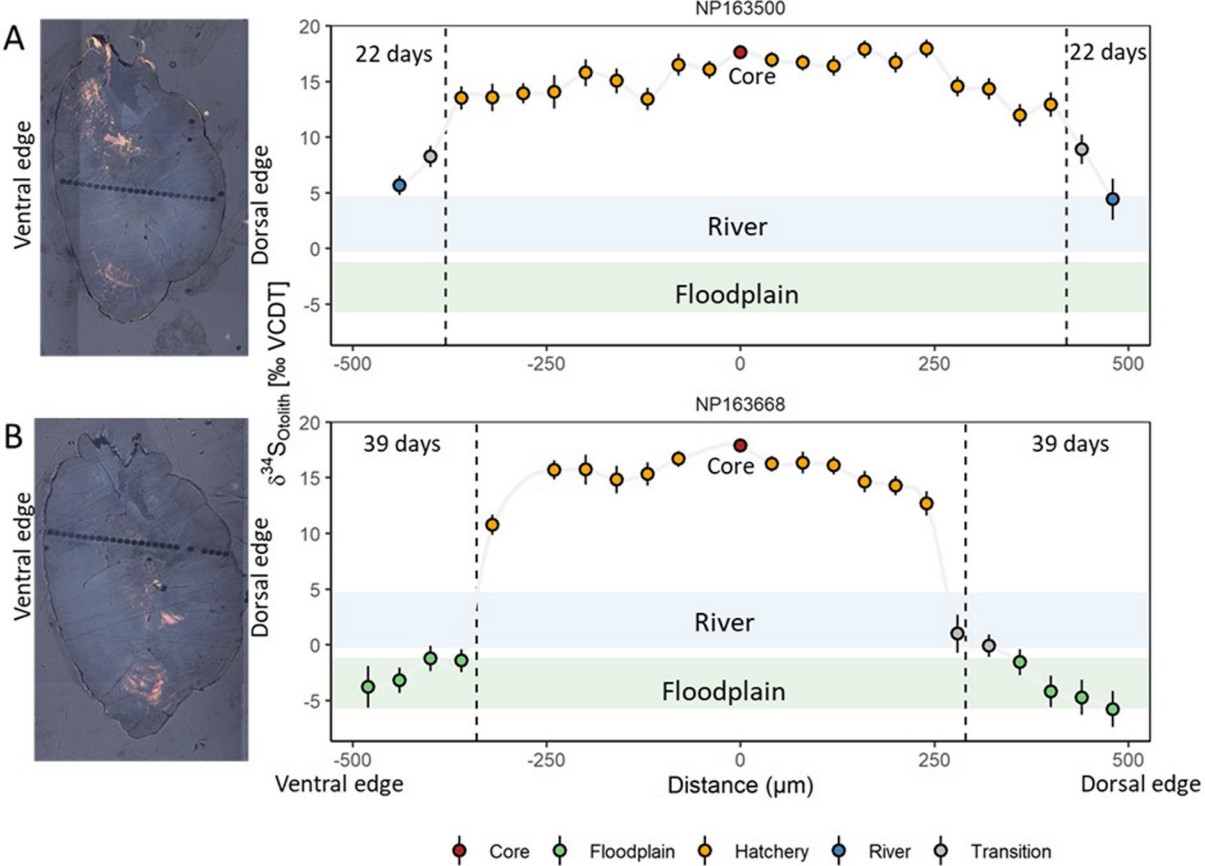

**Fig 7.** Example $\delta^{34}S_{Otolith}$ profiles for a fish reared on the river (A) and floodplain (B). Spots that overlapped the timing of movement from the hatchery to the river or floodplain were classified as "transition" and not included in the calculation for the $\delta^{34}S_{Otolith}$ means as they represent a mixture of otolith growth in different habitats.

floodplain food webs, demonstrating the sensitivity of isotopic turnover rates to growth rate in natural systems [16]. Body size is important for juvenile Chinook Salmon because size has been associated with increased juvenile survival in the ocean, especially in years of poor ocean conditions [27, 79]. Freshwater habitat could become increasingly important if ocean conditions worsen due to climate change [80, 81].

If salmon reared for less than 30 days on the Yolo Bypass, our tissue assimilation work suggests salmon muscle may not reach the $\delta^{34}S$ and $\delta^{13}C$ values of the prey on the floodplain. Yet, the mixing models detailed in Moore et al., (2016) demonstrate how understanding the assimilation rates of $\delta^{13}C$ and $\delta^{34}S$ in muscle as well as the habitat-specific $\delta^{13}C$ and $\delta^{34}S$ values can be used to estimate estuarine residency in juvenile salmon even when tissues are not in equilibrium [12, 16, 17]. Our study also provides the necessary data that can be used in future studies to estimate floodplain residency here in the CCV. Additionally, other soft tissues such as individual eye lens laminae that function as archival chronometers or livers and fins that have faster turnover rates will be valuable additions in future studies to capture isotopic shifts over shorter time intervals of floodplain rearing [9, 17, 21, 71, 82–84]. It is noteworthy, that in years where there was natural recruitment of juvenile Chinook Salmon on the floodplain, many of the fish captured leaving the floodplain displayed values in their stomach contents similar to those of the caged floodplain fish, indicating their use of the floodplain for foraging and not just as a migratory corridor.

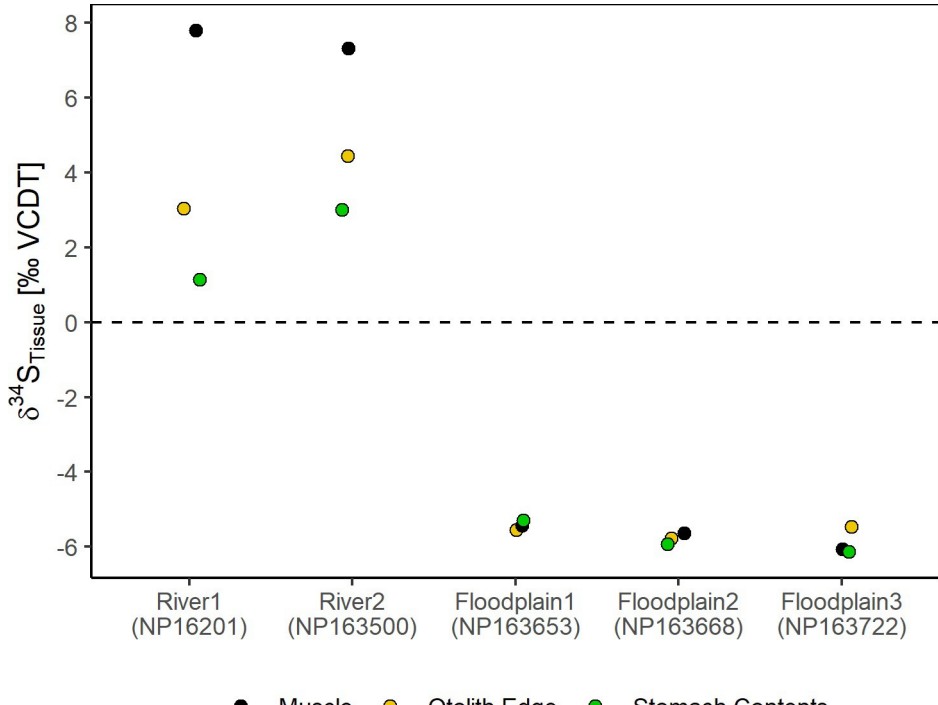

**Fig 8. Comparison of the otolith edge (30μm) $\delta^{34}S_{Otolith}$ values with the $\delta^{34}S_{muscle}$ and $\delta^{34}S_{stomach}$ values for two river and three floodplain fish.**

Sulfur isotopes in juvenile otoliths reflected dietary differences in $\delta^{34}S$ values among hatchery feed, prey on the floodplain and in the river. Unlike muscle tissue isotopes that reach equilibrium within ~30 days on the floodplain, in situ measurements of $\delta^{34}S$ in otoliths allow for finer temporal resolution (days to weeks) of floodplain rearing. In Chinook Salmon, the core of the otolith reflects a marine $\delta^{34}S$ value, inherited from the $\delta^{34}S$ value from the mother during egg formation. After emergence and start of feeding, $\delta^{34}S$ values should reflect the dietary input over that otolith growth period. The otolith $\delta^{34}S$ data showed a diet shift from hatchery to river or floodplain environment, indicating that this dietary isotopic signal is permanently archived in fish. It is noteworthy that for wild-caught river fish, the stomach contents, muscle, and $\delta^{34}S$ otolith values were not in equilibrium (Fig 8). This could be due to temporal variability in diets or an artifact of the small sample size. Future research characterizing temporal variability in in-river salmon diets in coordination with expanded analysis of more otoliths would be helpful to understand factors influencing these observed discrepancies. In our study we did not measure $\delta^{13}C$ in the otoliths, which are complicated to interpret due to the incorporation of both organic and non-dietary sources of carbon to the carbonate structure [83]. Our study demonstrates that otoliths can be used to retrospectively quantify habitat use and duration of time adult Chinook Salmon spent on the floodplain even if it is during a short flood event that is less than 30 days.

Fish captured in the Sacramento River did not have access to the Yolo Bypass and displayed a wide range of $\delta^{34}S$ and $\delta^{13}C$ values in their stomach contents. Nevertheless, the LDA analyses were able to classify these fish into the correct habitat with high accuracy. However, the variation in $\delta^{34}S$ value of the stomach contents from the wild-caught salmon in the river suggests that these individuals may have reared in other off-channel habitats with similar biogeochemical properties to the Yolo Bypass floodplain. One potential habitat these fish may have used is

the Sutter Bypass, located upstream of the Yolo Bypass and which is inundated more frequently, including some drought years [47]. The salmon reared in cages within the Sacramento River showed less variability and consistently higher $\delta^{34}S$ values within their stomach contents compared to floodplain-reared fish. The otoliths from fish reared in cages within the Sacramento River permanently archived these food web differences with higher $\delta^{34}S$ values compared to those on the floodplain. Given the potential for large river systems to have multiple floodplains or habitats that may create anaerobic hyporheic environments, holistic sampling of these habitats will inform the scale of inference for the isotopic markers. For example, if multiple floodplains had lower $\delta^{13}C$ and $\delta^{34}S$ values and overlapping values, then additional tools would be required to differentiate contributions of specific floodplains. In the CCV, $^{87}Sr/$ $^{86}Sr$ in otoliths have been used to identify natal sources for juvenile salmon based on differences in watershed geology and water chemistry [85–88]. Therefore, the inclusion of multiple isotopes such as $\delta^{13}C$ and $\delta^{34}S$ that captures differences in food webs between floodplain and riverine habitats in coordination with otoliths $^{87}Sr/$ $^{86}Sr$ that reflects different water sources could provide finer spatial resolution of salmon habitat use at a landscape scale.

The value in using $\delta^{34}S$ in addition to $\delta^{13}C$ to distinguish between riverine floodplains from other wetlands is the potential for $\delta^{13}C$ values to overlap in these areas that see detrital inputs from the decomposition of plants and other organic matter [40, 42]. A natural gradient exists from fresh water-to-salt water, with $\delta^{34}S$ increasing, as salinity increases, suggesting $\delta^{34}S$ could also be a valuable tool to distinguish between fluvial floodplains from tidal or estuarine wetlands [43]. Additionally, $\delta^{13}C$ and $\delta^{15}N$ fractionate, increasing as they move up trophic levels ($\delta^{13}C = + 1$‰; $\delta^{15}N = + 3$–4‰), whereas $\delta^{34}S$ exhibits little to no fractionation as it moves through the food web [89]. Our results also demonstrated why $\delta^{15}N$, an isotope commonly paired with $\delta^{13}C$, was not useful to differentiate between floodplain and river habitat use in this system; the invertebrate food of juvenile salmon in each habitat tends to occupy the same trophic level, resulting in overlapping $\delta^{15}N$ values.

Our study included years that spanned climatic extremes of drought (2012–2015) [61] and flood conditions, (1999 and 2017) as well as one year that was average with moderate flooding (2016). These climatic patterns provided an opportunity to understand the extent to which consistent biogeochemical processes occurred on floodplains under different hydrologic and environmental conditions. Study years with flooding also gave us the opportunity to study juvenile Chinook Salmon that naturally recruited to the floodplain. While prey resources and diet composition have been shown to vary between drought versus flood conditions [31, 32], taxonomic variation in stomach contents of floodplain fish was consistent among years. Across all years, there appeared to be a distinct difference in invertebrate prey items available to the juvenile Chinook Salmon between habitats. Other studies done in the same region, studied these taxonomic differences between floodplain and river habitats in greater detail with similar results [27, 29, 35, 43, 85].

While the $\delta^{34}S$ values seen in the stomach contents and muscle tissue of floodplain reared fish varied among years, they were consistently lower compared to fish collected in the river. Interestingly, the lowest isotopic values occurred during drought years, presumably because increased temperatures, lower flows, and higher residence time of the water on the floodplain may have caused more sulfur reduction and plant decomposition by bacteria than in higher flow years [89, 90]. But even during the wettest year on record (2017), the Yolo Bypass floodplain food web was still characterized by $\delta^{34}S$ values less than 0 ‰ for both enclosed/caged and most of the naturally recruited floodplain fish captured, demonstrating how $\delta^{34}S$ is a robust and consistent measure of this floodplain habitat in all water years.

While many of the fish used in this study were reared in a small, managed region of northern Yolo Bypass, their $\delta^{34}S$ values were generally consistent with fish that naturally recruited and that

were captured from locations throughout the floodplain. The results from the mixed effects model highlighted the habitat heterogeneity of both $\delta^{13}C$ and $\delta^{34}S$ isotopic values found in prey items of fish captured or reared throughout the Yolo Bypass. The Yolo Bypass is 23,867 hectares of diverse agriculture and wildlife habitat. Both $\delta^{13}C$ or $\delta^{34}S$ are isotopes where fractionation can be caused by decomposition of plant material. It is not surprising then that these two isotope values would not be uniform across such a heterogeneous landscape [31, 89], but they were still distinct from the adjacent Sacramento River. Previous research from the Yolo bypass has found carbon to be primarily sourced through heterotrophic pathways [91]. This is consistent in our study where $\delta^{13}C$ and $\delta^{34}S$ values in the prey items of fish captured throughout the Yolo Bypass were consistently lower than those captured in the Sacramento River for all years sampled.

Biogeochemical processes on the Yolo Bypass floodplain created a distinctive habitat marker; lower $\delta^{13}C$ and $\delta^{34}S$ values were found in salmon diets, remained remarkably distinct over time and across variable hydrologic conditions, and were incorporated into muscle tissues, and otoliths. While this study featured a single floodplain and river system, the biogeochemical processes that occurred within the Yolo Bypass and the Sacramento-San Joaquin watershed are not unique. These processes are a common feature of wetlands and floodplain systems where benthic food webs are fueled by the decomposition of plant material and other organic matter through anaerobic metabolic pathways [9, 35, 37, 39]. This allows for isotopes like $\delta^{13}C$ and $\delta^{34}S$ to be reduced by methane-oxidizing bacteria and sulfate-reducing bacteria respectively, resulting in lower values than those found in non-wetland habitats. For this reason, diet-based isotope markers show promise in nursery habitats where biogeochemical processes can lead to predictable food web communities with distinctive isotopic compositions.

Demonstrating habitat differences in $\delta^{13}C$ and $\delta^{34}S$ that are stable over time is a key first step in the development of isotope tools to identify floodplain rearing to address several ecological and management applications. The results of this study show promise for better understanding the ontogenetic patterns of habitat use in salmon. Recent advancements in the use of stable isotopes in individual eye lens laminae to reconstruct floodplain use for salmon in the CCV provides a feasible method for broad-scale ecological and fisheries applications [71].This approach opens avenues for quantitative analyses to reconstruct juvenile floodplain habitat use in juveniles downstream of the floodplain, adults in the fishery, or returning spawners. These estimates are required to understand and quantify the role of floodplains as salmon nursery habitats and their potential function in salmon population dynamics. $\delta^{34}S$ measurements in otoliths offers a finer temporal scale of floodplain use over eye lenses and allows for quantification of the number of days salmon access floodplains. Application of diet-based stable isotope tracers across multiple tissues, including salmon eye lenses and otoliths, can help guide future restoration and recovery for California salmon and other native fishes that rely on river-floodplain systems.

## Supporting information

**S1 File.**
(DOCX)

## Acknowledgments

George Whitman, Amber Manfree, Jamie Sweeney, Mollie Ogaz, Dana Myers, Kelly Neal, Anna Sturrock, Veronica Corbett, Eric Holmes, Rosa Cox, Emma Cox, UC Davis Stable Isotope Facility, Delta Juvenile Fish Monitoring Program, and California Department of Fish and Wildlife provided invaluable project assistance and sample collection. We also thank our anonymous reviewers who greatly improved this paper through the review process.

## Author Contributions

**Conceptualization:** Miranda Bell-Tilcock, Carson A. Jeffres, Jacob V. E. Katz, Ted R. Sommer, J. Louise Conrad, Rachel C. Johnson.

**Data curation:** Miranda Bell-Tilcock, Carson A. Jeffres, Malte Willmes, Richard A. Armstrong, Peter Holden, Jacob V. E. Katz, Ted R. Sommer, J. Louise Conrad, Rachel C. Johnson.

**Formal analysis:** Miranda Bell-Tilcock, Carson A. Jeffres, Andrew L. Rypel, Malte Willmes, Rachel C. Johnson.

**Funding acquisition:** Carson A. Jeffres, Jacob V. E. Katz, Ted R. Sommer, Rachel C. Johnson.

**Investigation:** Miranda Bell-Tilcock, Carson A. Jeffres, J. Louise Conrad, Rachel C. Johnson.

**Methodology:** Miranda Bell-Tilcock, Carson A. Jeffres, Peter Holden, Jacob V. E. Katz, J. Louise Conrad, Rachel C. Johnson.

**Project administration:** Carson A. Jeffres, Rachel C. Johnson.

**Resources:** Richard A. Armstrong.

**Supervision:** Carson A. Jeffres, Andrew L. Rypel, Peter B. Moyle, Nann A. Fangue.

**Visualization:** Miranda Bell-Tilcock, Andrew L. Rypel, Rachel C. Johnson.

**Writing – original draft:** Miranda Bell-Tilcock, Carson A. Jeffres, Andrew L. Rypel, Malte Willmes, Peter B. Moyle, Nann A. Fangue, Jacob V. E. Katz, Ted R. Sommer, J. Louise Conrad, Rachel C. Johnson.

**Writing – review & editing:** Miranda Bell-Tilcock, Carson A. Jeffres, Andrew L. Rypel, Malte Willmes, Richard A. Armstrong, Peter Holden, Peter B. Moyle, Nann A. Fangue, Jacob V. E. Katz, Ted R. Sommer, J. Louise Conrad, Rachel C. Johnson.

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
