## [Decision Letter · Decision Letter 0]

16 Sep 2020

PONE-D-20-24634

Biogeochemical processes create distinct isotopic fingerprints to track floodplain rearing of juvenile salmon

PLOS ONE

Dear Ms. Bell-Tilcock,

Thank you for submitting your manuscript to PLOS ONE. I have received four reviews of the manuscript. In all cases, the reviewers were positive about the scientific contribution your work will make and three of the four reviewers recommended that the manuscript would be suitable for acceptance after only minor revisions. However, after careful consideration, I think the detailed and extensive comments of reviewer 2 (contained in the attachment) raise a number of interesting and important questions, particularly with respect to how the data are handled statistically.  So I conclude that the manuscript has merit but does not fully meet PLOS ONE’s publication criteria as it currently stands. Therefore, we invite you to submit a revised version of the manuscript that addresses the points raised during the review process.

We look forward to receiving your revised manuscript.

Kind regards,

Lee W Cooper, Ph.D.

Biogeochemistry Section Editor

PLOS ONE

Journal Requirements:

3. Please expand the acronym “DWR” (as stated in your financial disclosure) so that it states the name of your funders in full.

5. We note that Figure 1 in your submission contains map images which may be copyrighted.

We require you to either (a) present written permission from the copyright holder to publish these figure specifically under the CC BY 4.0 license, or (b) remove the figure from your submission:

b. If you are unable to obtain permission from the original copyright holder to publish these figure under the CC BY 4.0 license or if the copyright holder’s requirements are incompatible with the CC BY 4.0 license, please either i) remove the figure or ii) supply a replacement figure that complies with the CC BY 4.0 license. Please check copyright information on all replacement figures and update the figure caption with source information. If applicable, please specify in the figure caption text when a figure is similar but not identical to the original image and is therefore for illustrative purposes only.

Reviewers' comments:

Reviewer's Responses to Questions

**Comments to the Author**

1. Is the manuscript technically sound, and do the data support the conclusions?

Reviewer #1: Yes

Reviewer #2: Partly

Reviewer #3: Yes

Reviewer #4: Yes

2. Has the statistical analysis been performed appropriately and rigorously? 

Reviewer #1: Yes

Reviewer #2: No

Reviewer #3: Yes

Reviewer #4: Yes

3. Have the authors made all data underlying the findings in their manuscript fully available?

Reviewer #1: Yes

Reviewer #2: Yes

Reviewer #3: Yes

Reviewer #4: Yes

4. Is the manuscript presented in an intelligible fashion and written in standard English?

Reviewer #1: Yes

Reviewer #2: Yes

Reviewer #3: Yes

Reviewer #4: Yes

5. Review Comments to the Author

Reviewer #1: Review summary

This manuscript employed carbon and sulfur stable isotopes as a marker to evaluate habitat utilization of juvenile salmon in a large floodplain. Through large-scale field experiments, the authors found significant differences in stomach content and muscle tissue d13C and d34S between floodplain reared fish and those collected in adjacent rivers. Tissue assimilation experiments confirmed that sulfur stable isotopic composition of the diet was reflected in the muscle tissue after several weeks. Differences in stable isotopic composition between the two habitats were stable across years of different environmental conditions. Additionally, otolith profile analysis of d34S corresponded well with shifting diet and habitat use of juvenile salmon. These results suggest that sulfur isotopes could be widely used as a promising and powerful tool for evaluating habitat use and connectivity of fish species in large river systems. Overall, this is an intriguing study that has important implications for habitat restoration and management. I believe that the topic is appropriate for the readers of the journal and I recommend publication with minor edits. Please see below for specific suggestions.

Minor Comments

L92: For describing isotope values, authors should consider using high/low instead of enriched/depleted because d13C are numbers and they cannot be depleted or enriched. This expression should be reflected throughout the manuscript.

L366: Fig. “3” instead of Fig. “2”

L387: Fig. “3” instead of Fig. “2”. Please also indicate what the ellipses in the figure represent.

L420: Please indicate the alpha value used for statistical significance in the methods because water year type (P = 0.019) could become significant with an alpha value of 0.05

L431: Fig. “6” instead of Fig. “5”

L442: Please indicate what the solid lines represents in the figure.

Reviewer #2: 1. The claim that sulfur met all the authors criteria as a floodplain marker and can be used in a retrospective analysis of floodplain use is not supported by the data. The data clearly indicate that fish rearing less than 39 days on the floodplain would be classified as river-reared fish. This means that the ability of S to differentiate habitats is dependent on rearing time which would hamstring a retrospective analysis for years when inundation is less than 39 days.

2. The statistical analysis treats all fish samples as equivalent however, their "rearing type" is variable (caged, experimental pond, volitional migrating). Figures 4 and 5 suggest that rearing type influences isotopic composition. Since the data are unbalanced in regards to fixed effects, an analysis where these confounding influences are accounted for is warranted. Alternatively, analyses could be performed with subsets of the data that are more comparable.

Reviewer #3: In this manuscript, the authors use stable isotopes to track floodplain rearing in California chinook salmon. The analysis appears sound, and the conclusions are supported by the datasets provided. Below I provide a number of items for the authors to consider

Major

1) Introduction would be strengthened by doing some re-organizing. First paragraph is great, starts with the broad, global importance of wetlands, but then jumps into salmon without much of a link. I would suggest splitting the first paragraph at line 59, and then adding some detail onto the salmon part.

The rest of the introduction also seems to go back and forth between specific applications of SIA to salmon in California, and broader development of methods and approaches to understand migratory history of animals. I think a bit of restructuring could help to go from the broadest ideas (i.e. wetlands are important, but hard to track importance) to narrower (i.e. SIA can help) to narrowest (i.e. SIA can help understand dynamics of California Chinook).

2) L266-267: From the text here, it’s not super clear how stomach content differences were tested/modelled? Were all stomachs pooled from floodplain and river? Across all years? Sites? Fig 1 looks like everything was pooled, but L548-550 make it sound like annual comparisons were made

3) L494-495: The Woodson reference is interesting because it appears that size is most important in years of low recruitment, perhaps this is worth highlighting here (especially since it looks like ocean conditions are not getting more favourable for California Chinook (e.g. Wells et al. 2016))

Wells, B. K., Santora, J. A., Schroeder, I. D., Mantua, N., Sydeman, W. J., Huff, D. D., & Field, J. C. (2016). Marine ecosystem perspectives on Chinook salmon recruitment: a synthesis of empirical and modeling studies from a California upwelling system. Marine Ecology Progress Series, 552, 271-284.

Minor

L 98: Maybe reword to “d34S is less often used in the isotope ecology literature”

L150 remove “during their early ocean years”

L234 – Fig 2 could go in supplement

L 302 – list the random effects here again

L302 – might be worth including a sentence on why you thought these random effects might influence assimilation rates of d34S.

L418-420 – if this is still not statistically significant, is it worth pointing out?

L475 – 487 – I would remove the references from Oele et al from this paragraph and put them later on. Good material, but I think this paragraph is stronger if you just summarize your own work

L511-513: I would add a “potentially” here. This is likely true, but this paper doesn’t use adult otoliths, so this is speculative

L585-598: Remove references to eye lenses here, not covered in this paper

Reviewer #4: This ms might be better off as several separate ms enabling greater discussion of each of the various approaches. Would have liked to see the C-13 & N-15 data for the amphipods to match the fig in app A. IMHO the most problematic aspect in using S-34 is that there are 3 endmembers - one the very enriched marine value, two the slightly negative reduced value seen in Yolo and three the slightly positive "normal" freshwater endmember. This latter one seems to vary by location with First Mallard and Lindsey being the most positive. Why was this so? Yolo is not quite as negative in S-34 in the wet year - is this evidence of a nutrient subsidy from the main river? A bit disappointed in only N=5 otoliths in two categories so only N=2 and N=3 per group!!!

6. PLOS authors have the option to publish the peer review history of their article (what does this mean?). If published, this will include your full peer review and any attached files.

Reviewer #1: No

Reviewer #2: No

Reviewer #3: No

Reviewer #4: No

---

## [Author Response · Author response to Decision Letter 0]

8 Jan 2021

Response to reviewer comments can be found in the letter to the editor as well as a separate file I uploaded entitled "Response to reviewer comments".

---

## [Decision Letter · Decision Letter 1]

28 Mar 2021

PONE-D-20-24634R1

Biogeochemical processes create distinct isotopic fingerprints to track floodplain rearing of juvenile salmon

PLOS ONE

Dear Dr. Jeffres,

Thank you for submitting your revised manuscript to PLOS ONE. I asked Reviewer #2 to evaluate the revised manuscript and determine if there were additional revisions that might improve the scientific value of the contribution. Based upon the constructive comments made by this reviewer with respect to the revised version, I invite you to revise and re-submit a new version of the manuscript that addresses the points raised during this review cycle, particularly a better description of the mixed effects model used and putting the statistical value of the otolith data into better context. The specific recommendations that Reviewer #2 makes are appended below.

I look forward to receiving your revised manuscript; thank you again for submitting your contribution to PLOS ONE.

Kind regards,

Lee W Cooper, Ph.D.

Academic Editor

PLOS ONE

Journal Requirements:

Reviewers' comments:

Reviewer's Responses to Questions

**Comments to the Author**

1. If the authors have adequately addressed your comments raised in a previous round of review and you feel that this manuscript is now acceptable for publication, you may indicate that here to bypass the “Comments to the Author” section, enter your conflict of interest statement in the “Confidential to Editor” section, and submit your "Accept" recommendation.

Reviewer #2: (No Response)

2. Is the manuscript technically sound, and do the data support the conclusions?

Reviewer #2: Yes

3. Has the statistical analysis been performed appropriately and rigorously? 

Reviewer #2: Yes

4. Have the authors made all data underlying the findings in their manuscript fully available?

Reviewer #2: Yes

5. Is the manuscript presented in an intelligible fashion and written in standard English?

Reviewer #2: Yes

6. Review Comments to the Author

Reviewer #2: The authors did a nice job of addressing the previous comments and this version of the manuscript is significantly improved. There is still an issue with a lack of clarity in the description of methods and results for the mixed effect model (described below) that should be addressed. Additionally, there are some important features of the study that could influence interpretation of the findings that should be addressed specifically in the discussion including the caging effect on the Sacramento River fish and the low number of otoliths examined.

Mixed model

The authors put a lot of effort into revising the mixed effect models to account for sources of variation in the data. However, the description of the methods and results are still unclear. For example, on lines 270-271 the model is described as applied to stomach contents, but supplemental Table 1 has model results for both stomach contents and muscle tissue. In the main text, there are no mixed effect model results for muscle tissue, only a description of the boxplot (although the discussion mentions muscle tissue on lines 562-564). Related to this is the LDFA analysis that was applied to stomach contents but not muscle tissue. If both sources of information are available (stomach contents and muscle), why not apply the same statistical models to both? Doing this would provide a nice way to determine how well the chosen stable isotopes can classify rearing location for different assimilation periods (stomach contents=source, fish=time integrated assimilation, otolith=permanent record). If muscle tissue results are not analyzed and reported in the same way as the stomach contents, some justification for excluding it should be provided.

The description of the mixed model results on lines 379-386 is insufficient. The authors describe that all sulfur values in the Yolo Bypass were significantly lower than those in the river but no test statistics or coefficients are provided. Supplementary Table 1 does not provide any test statistics or coefficients for the river floodplain comparison or a description of which model was most parsimonious according to the LR tests. Providing these is important to be able to interpret the results, particularly if findings of significant differences are asserted.

Line279-283: “Caged” was not included as a random effect but were those fish included in the model?

Discussion

There is a lot of set up in the introduction about the value of otoliths but then only 5 total otoliths (three floodplain and two river) are analyzed with only two of the five reported in the main text. Some discussion of the uncertainty inherent in this low sample size is needed. The caged river fish that otoliths were sampled from had much less trophic variability relative volitionally migrating fish and floodplain fish. Additionally, the variation in sulfur isotope ratios between the river and floodplain was reduced in wet years. Both of these factors could reduce clarity in the archival record of otoliths (and time integrated value of muscle tissue) under natural rather than treatment conditions. The data presented suggests the use of stable isotopes of sulfur to identify floodplain rearing is very promising. An explicit discussion of some of the study caveats and how they might be addressed in future studies would be a nice addition.

7. PLOS authors have the option to publish the peer review history of their article (what does this mean?). If published, this will include your full peer review and any attached files.

Reviewer #2: No

---

## [Author Response · Author response to Decision Letter 1]

30 Aug 2021

Reviewer #2: 

The authors did a nice job of addressing the previous comments and this version of the manuscript is significantly improved. There is still an issue with a lack of clarity in the description of methods and results for the mixed effect model (described below) that should be addressed. Additionally, there are some important features of the study that could influence interpretation of the findings that should be addressed specifically in the discussion including the caging effect on the Sacramento River fish and the low number of otoliths examined.

COMMENT: Mixed model

The authors put a lot of effort into revising the mixed effect models to account for sources of variation in the data. However, the description of the methods and results are still unclear. For example, on lines 270-271 the model is described as applied to stomach contents, but supplemental Table 1 has model results for both stomach contents and muscle tissue. In the main text, there are no mixed effect model results for muscle tissue, only a description of the boxplot (although the discussion mentions muscle tissue on lines 562-564). 

Related to this is the LDFA analysis that was applied to stomach contents but not muscle tissue. If both sources of information are available (stomach contents and muscle), why not apply the same statistical models to both? Doing this would provide a nice way to determine how well the chosen stable isotopes can classify rearing location for different assimilation periods (stomach contents=source, fish=time integrated assimilation, otolith=permanent record). If muscle tissue results are not analyzed and reported in the same way as the stomach contents, some justification for excluding it should be provided.

RESPONSE:

We have clarified language in this section. Namely, all the muscle values have been removed from the mixed effects model (MEM) in the supplemental (Table 1) and removed from the corresponding text. The stated objective of the MEM was to ask the extent to which the CNS values varied between habitats as a function of water year. The diet content of the individual fish is the best source of information for that analysis as it represents the proximate source-diets on the two habitats. The muscle data from the paired fish includes variability due to time spent in the habitats and time integrated assimilation and was therefore deemed not appropriate for the MEM. We have crystalized the data sources and specific statistical analyses used to answer the specific objectives in the first paragraph of the results as a roadmap for the results based on the helpful framework of (stomach contents=source, fish=time integrated assimilation, otolith=permanent record). Due to the different assimilates rates of the different tissues, specific data sources were used to address the individual questions. 

COMMENT: 

The description of the mixed model results on lines 379-386 is insufficient. The authors describe that all sulfur values in the Yolo Bypass were significantly lower than those in the river but no test statistics or coefficients are provided. Supplementary Table 1 does not provide any test statistics or coefficients for the river floodplain comparison or a description of which model was most parsimonious according to the LR tests. Providing these is important to be able to interpret the results, particularly if findings of significant differences are asserted.

Line279-283: “Caged” was not included as a random effect but were those fish included in the model?

RESPONSE:

We have clarified language in this section and added statistical analysis describing differences in spatial variation in δ¹³C and δ³⁴S. Test statistics are now included in the text (lines 845-901) in the results sections and Supplemental Table 1 has been simplified. 

COMMENT:

Discussion

There is a lot of set up in the introduction about the value of otoliths but then only 5 total otoliths (three floodplain and two river) are analyzed with only two of the five reported in the main text. Some discussion of the uncertainty inherent in this low sample size is needed. The caged river fish that otoliths were sampled from had much less trophic variability relative volitionally migrating fish and floodplain fish. Additionally, the variation in sulfur isotope ratios between the river and floodplain was reduced in wet years. Both of these factors could reduce clarity in the archival record of otoliths (and time integrated value of muscle tissue) under natural rather than treatment conditions. The data presented suggests the use of stable isotopes of sulfur to identify floodplain rearing is very promising. An explicit discussion of some of the study caveats and how they might be addressed in future studies would be a nice addition.

RESPONSE:

Sulfur isotopes in otoliths remains more of a boutique assay and remains incredibly time consuming and expensive for ecological/fisheries samples sizes (although see Johnson et al. 2012). The expense of this assay and limited project budget precluded a larger numbers of otoliths to be analyzed and therefore the inclusion of otoliths served more of a proof of concept. This is highlighted in the figure of the two otoliths as “examples” of the two distinct profiles. However, all 5 otoliths were analyzed and discussed in the paper. We have clarified this in the paper. We recognize that as with any small sample size, one risks the samples selected not representing the entire variability exhibited in a population. We have added this as a limitation of the study and suggest further research needed to adequately capture the individual variability that may exist, especially given the variability observed in the 2 wild-caught fish. Since the submission of this manuscript, a complementary paper on the use of fish eye lens isotopes has been published (Bell-Tilcock et al. 2020) and is a much more appropriate tissue to characterize floodplain use as an archival tissue (as lenses are more protein-rich than otoliths) and the analytical throughput is significantly faster as laminae are submitted to traditional stable isotope facilities for analysis. The benefit of the otoliths is that one could link the microstructure to the geochemistry and interpret the number of days fish reared on flooplains whereas the specific time-chronology recorded in an individual fish lens laminae remains uncalibrated but more feasible for ecological and fisheries application. We updated the discussion to include this new research and a eye lens/otolith comparison.

---

## [Editor Report · Decision Letter 2]

2 Sep 2021

Biogeochemical processes create distinct isotopic fingerprints to track floodplain rearing of juvenile salmon

PONE-D-20-24634R2

Dear Dr. Jeffres,

We’re pleased to inform you that your manuscript has been judged scientifically suitable for publication and will be formally accepted for publication once it meets all outstanding technical requirements. I appreciate the efforts you and your co-authors made to respond to the reviewer comments and I judge that this final round of revisions has been sufficient to meet the final recommendations of Reviewer #2. A couple of minor points---I would change the discussion of the sulfur isotope standard by referring to Vienna-Canyon Diablo rather than Vienna Canyon Diablo. The Vienna prefix just refers to the involvement of the International Atomic Energy Agency (in Vienna) in the development of isotope standards, but the Canyon Diablo name was the original standard relating to the meteorite fragments from Meteor Crater in Arizona that turned out to be not sufficiently homogeneous to be suitable as an international standard. You might also consider thanking, in the acknowledgments, the three anonymous reviewers who helped improve the manuscript through the review process.

As far as next steps, within one week, you’ll receive an e-mail detailing the required amendments. When these have been addressed, you’ll receive a formal acceptance letter and your manuscript will be scheduled for publication.

If your institution or institutions have a press office, please notify them about your upcoming paper to help maximize its impact. Given the public interest in the continued sustainability of salmon populations in California's Central Valley, I think your findings and your innovative use of sulfur isotopes could be of public interest if explained appropriately, especially in light of the critical water management needs now being addressed to balance agriculture and fisheries management in California. If your public affairs office will be preparing press materials, please inform our press team as soon as possible -- no later than 48 hours after receiving the formal acceptance. Your manuscript will remain under strict press embargo until 2 pm Eastern Time on the date of publication. For more information, please contact onepress@plos.org.

Kind regards,

Lee W Cooper, Ph.D.

Section Editor

PLOS ONE

---

## [Editor Report · Acceptance letter]

19 Oct 2021

PONE-D-20-24634R2 

Biogeochemical processes create distinct isotopic fingerprints to track floodplain rearing of juvenile salmon 

Dear Dr. Jeffres:

I'm pleased to inform you that your manuscript has been deemed suitable for publication in PLOS ONE. Congratulations! Your manuscript is now with our production department. 

Kind regards, 

on behalf of

Dr. Lee W Cooper 

Section Editor

PLOS ONE